# OpenReview forum: "Can LLM-Generated Misinformation Be Detected?"
_ICLR.cc/2024/Conference — ICLR 2024 poster_

### Official Review · Reviewer_NKW5 · 2023-10-19

**Soundness:** 2 fair
**Presentation:** 2 fair
**Contribution:** 2 fair
**Rating:** 5
**Confidence:** 4

**Summary:**

The paper investigates the possibility of the generation of misinformation from LLMs, whether human evaluators can identify LLM-generated misinformation, and assesses the performance of automated detectors in identifying human vs. LLM-generated misinformation. Among other things, the paper finds that LLM-generated misinformation is harder to detect for humans compared to human-written misinformation and that LLM-generated misinformation is harder to detect for automated methods.

**Strengths:**

I would like to thank and applaud the authors for working on this very important and timely problem; this research has the potential to have a big impact on the research community and help mitigate emerging problems like the spread of misinformation online. I liked that the paper creates a taxonomy of LLM-generated misinformation and then goes further and investigates the generation of LLM misinformation across types, sources, and domains. Also, an important strength of the paper is that the paper’s evaluation considers many state-of-the-art LLMs used for in-context learning purposes to solve the problem of misinformation detection.

**Weaknesses:**

While I believe that this study is an important research effort, I have some concerns with the way that the paper conducts the experiments and the interpretation of the results. Below, I provide more details on my main concerns with the paper.

First, the paper’s evaluation is done on a very small scale, particularly 100 pieces of news, and leveraging only 10 human evaluators to assess the performance of humans and compare it with various LLM-based automated detectors. Due to this, I am wondering how robust and generalizable the presented results are. At the same time, the paper does not discuss whether the presented results and differences between human-written and LLM-generated misinformation are statistically significant. That is, the paper simply presents the results and differences without providing any additional context of how statistically significant the results. I suggest to the authors to consider expanding their evaluation and discussing the statistical significance of these results.

Second, the paper lacks important details on how the ten human evaluators are selected. Do these evaluators have previous experience with annotating piece of information as misinformation or not? Do you take any steps to ensure that the annotations are of high quality and that the annotators did not use LLM to solve the task? For instance, the paper by Veselovsky et al. [1] demonstrated that crowd workers are using LLMs to solve tasks, so I am wondering if the paper took any steps to ensure that the human evaluators solved the task on their own. I think this is a crucial part of the paper as many results rely on the quality of these annotations and more details can shed light into these concerns. Finally, it is unclear to me why the paper studies the performance of the human evaluations on a per evaluator basis rather than taking the majority agreement of the evaluators per piece of information and then reporting the results on aggregate. Also, I suggest to the authors to include the inter-annotator agreement of the evaluators so that we can assess how difficult was the presented task for them.

Finally, from the paper, it’s unclear how the attack rates in Section 3 are calculated. Are these based on manual evaluations from the authors? I suggest to the authors to provide more details on how the annotated the generated pieces of information, how many people annotated each piece, etc.

Reference:

[1] Veselovsky, V., Ribeiro, M.H. and West, R., 2023. Artificial Artificial Artificial Intelligence: Crowd Workers Widely Use Large Language Models for Text Production Tasks. arXiv preprint arXiv:2306.07899.

**Questions:**

1. How are the 10 human evaluators selected, and did you take any steps to ensure that their annotations are of high quality? Also, did you take any steps to assess if the annotators used LLMs to solve the task (see paper by Veselovsky et al., 2023)
2. Are the presented results and differences between human and LLM-misinformation statistically significant?

---

> ### Author Response · Authors · 2023-11-23
> **Response to Reviewer NKW5**
>
> We greatly appreciate the feedback.
>
> 1. First, the paper’s evaluation is done on a very small scale, particularly 100 pieces of news, and leveraging only 10 human evaluators to assess the performance of humans and compare it with various LLM-based automated detectors. Due to this, I am wondering how robust and generalizable the presented results are.
>
>
> We conduct a much more extensive investigation on misinformation generated by ChatGPT, Llama-7b-chat-generated misinformation (or 13b, 70b) and Vicuna-7b-generated misinformation (or 13b, 33b) to further validate our findings. For each LLM as the misinformation generator, we compare the detection difficulty of different misinformation generation methods including Paraphrase Generation, Rewriting Generation, Open-ended Generation and human-written misinformation. We include multiple human-written misinformation datasets including Politifact, Gossipcop and CoAID. The results are in Appendix A.
>
> We will expand the human evaluation part in our future work.
>
> 2. At the same time, the paper does not discuss whether the presented results and differences between human-written and LLM-generated misinformation are statistically significant. That is, the paper simply presents the results and differences without providing any additional context of how statistically significant the results. I suggest to the authors to consider expanding their evaluation and discussing the statistical significance of these results.
>
>
> We carefully discuss the statistical significance of all the results in Appendix B.
>
>
>
> 3. Do these evaluators have previous experience with annotating piece of information as misinformation or not?
>
> We do not require the annotating experience. We ask evaluators to select a label of “factual” or “nonfactual” for each news item only based on their own first perceptions upon reading it. For each evaluator, the news articles are presented in a random order.
>
> 4.  Do you take any steps to ensure that the annotations are of high quality and that the annotators did not use LLM to solve the task? For instance, the paper by Veselovsky et al. [1] demonstrated that crowd workers are using LLMs to solve tasks, so I am wondering if the paper took any steps to ensure that the human evaluators solved the task on their own. I think this is a crucial part of the paper as many results rely on the quality of these annotations and more details can shed light into these concerns.
>
> Yes. We take some measures such as recording the annotating process to prevent the use of LLMs.
>
> 5. Finally, it is unclear to me why the paper studies the performance of the human evaluations on a per evaluator basis rather than taking the majority agreement of the evaluators per piece of information and then reporting the results on aggregate.
>
> Here we aim to compare the detection difficulty of LLM-generated misinformation & human-written misinformation. Thus, we need to control the human evaluators the same for both LLM-generated misinformation & human-written misinformation.
>
> 6. Finally, from the paper, it’s unclear how the attack rates in Section 3 are calculated. Are these based on manual evaluations from the authors? I suggest to the authors to provide more details on how the annotated the generated pieces of information, how many people annotated each piece, etc.
>
> Yes. We manually verify them. We will make it more clear in the revision.
>
> We hope we have addressed all your concerns. Thanks.

---

### Official Review · Reviewer_bQ63 · 2023-10-27

**Soundness:** 3 good
**Presentation:** 4 excellent
**Contribution:** 4 excellent
**Rating:** 8
**Confidence:** 5

**Summary:**

This paper studies a problem with high significance and urgency: detection of LLM-generated misinformation. More specifically, the development of advanced LLM make it easy for misinformation creators to efficiently generate misinformation. A critical question is: is the LLM-generated misinformation detectable? To understand this question better, the authors built up a LLM-generated misinformation dataset and then compare its detection difficulty with human-written misinformation for both human verifiers and machine learning models. Extensive experiments suggested that compared to human-written misinformation, LLM-generated misinformation is more deceptive and potentially more harmful.

**Strengths:**

1. Significance of the research question: AI-generated misinformation is a very critical problem for the development of LLM. The development of RLHF-based LLM can make the misinformation creators easily generate misinformation without any preliminary knowledge about deep learning. We urgently needed exploration on the topic.

2. Contribution to the community: This paper discuss the problem in a great details and can provide us with good resources (dataset and prompts) to study this problem.

3. Experiment details are discussed in details.

**Weaknesses:**

1. The dataset seems to be not very large. I understand that for evaluating human detection difficulty, we can not use too large dataset. But the authors can enlarge the dataset for evaluation of machine learning model.

2. For detector difficulty, the authors only discussed the zero-shot detection of generative LLMs. The results on other kinds of models (i.e. in-context-learning boosted LLMs, soft-prompt based LLMs, and encoder-based Large models like BERT and its variants) are not discussed.

3. Dataset is not opensourced. Actually, the data can be opensourced anonymously on GitHub.

**Questions:**

1. Will the datset be opensourced once the paper is accepted?

2. Is it possible to generate more data for detection evaluation?

3. Will few-shot learning improve the performance of the detection?

---

> ### Author Response · Authors · 2023-11-23
> **Response to Reviewer bQ63**
>
> We greatly appreciate the feedback.
>
> 1. The dataset seems to be not very large. I understand that for evaluating human detection difficulty, we can not use too large dataset. But the authors can enlarge the dataset for evaluation of machine learning model.
>
> We conduct a much more extensive investigation on misinformation generated by ChatGPT, Llama-7b-chat-generated misinformation (or 13b, 70b) and Vicuna-7b-generated misinformation (or 13b, 33b) to further validate our findings. For each LLM as the misinformation generator, we compare the detection difficulty of different misinformation generation methods including Paraphrase Generation, Rewriting Generation, Open-ended Generation and human-written misinformation. We include multiple human-written misinformation datasets including Politifact, Gossipcop and CoAID. The results are in Appendix A.
>
> 2. For detector difficulty, the authors only discussed the zero-shot detection of generative LLMs. The results on other kinds of models (i.e. in-context-learning boosted LLMs, soft-prompt based LLMs, and encoder-based Large models like BERT and its variants) are not discussed.
>
> We would like to explain why we adopt the zero-shot LLM detectors (e.g., GPT-4) to evaluate the detection difficulty of LLM-generated misinformation: (1) **The zero-shot setting can better reflect the real-world scenarios of detecting LLM-generated misinformation.** Conventional supervisedly trained detectors  (such as BERT in the pre-LLM era) are likely to be impractical to detect LLM-generated misinformation in the practices for the following two reasons. First, it is difficult to obtain factuality supervision labels to train detectors for LLM-generated misinformation since it is harder for humans to detect than human-written misinformation (Section 5). Second, malicious users can easily utilize methods shown in Table 1 and close-sourced LLMs (e.g., ChatGPT) or open-sourced LLMs (e.g., Llama2) to generate misinformation at scale in different domains, types, and errors, which is hard for conventional supervisedly trained detectors to maintain effective. (2) **Zero-shot Detectors can better reflect the intrinsic detection difficulty of  LLM-generated misinformation compared with human-written misinformation.** Conventional supervisedly trained detectors (such as BERT in the pre-LLM era) are strongly influenced by the labeled training set construction and cannot reflect the detection difficulty of test set well. For example, various factors in  the labeled training set construction (including adopting human-written or LLM-generated misinformation or their combination as the training set, mixing the misinformation data with different factual information data, the domains of training set, the percentage of misinformation data in the training set) can lead to completely different detection performances of human-written or LLM-generated misinformation test set. Zero-shot Detectors, which do not need labeled training set, can better reflect the intrinsic detection difficulty of human-written or LLM-generated misinformation test set. (3) **Zero-shot LLM detectors (e.g., GPT-4) are already almost the state-of-the-art misinformation detectors.** There are already many works that have shown directly prompting LLMs such as GPT-4 in a zero-shot way can outperform conventional supervisedly trained models such as BERT on detecting human-written misinformation [1,2,3,4,5]
>
> Thus, we adopt ChatGPT-3.5, GPT-4 and Llama2 with standard prompting and Chain-of-thought Prompting strategies as the **representative detectors** to evaluate the detection difficulty of LLM-generated misinformation data and also compare it with human-written misinformation data.
>
> [1] A Multitask, Multi-lingual, Multimodal Evaluation of ChatGPT on Reasoning, Hallucination, and Interactivity. https://arxiv.org/abs/2302.04023
>
> [2] Towards Reliable Misinformation Mitigation: Generalization, Uncertainty, and GPT-4 https://arxiv.org/abs/2305.14928
>
> [3] Interpretable Unified Language Checking https://arxiv.org/abs/2304.03728
>
> [4] Fact-Checking Complex Claims with Program-Guided Reasoning ACL 2023 https://aclanthology.org/2023.acl-long.386/
>
> [5] Assessing the Effectiveness of GPT-3 in Detecting False Political Statements: A Case Study on the LIAR Dataset https://arxiv.org/abs/2306.08190
>
> 3. Dataset is not opensourced. Actually, the data can be opensourced anonymously on GitHub.
>
> We release our whole datasets and detection results for reproduction and verification. https://anonymous.4open.science/r/llm-misinformation-iclr2024-7E35/
>
>
> We hope we have addressed all your concerns. Thanks.

---

### Official Review · Reviewer_oGY2 · 2023-11-01

**Soundness:** 1 poor
**Presentation:** 1 poor
**Contribution:** 2 fair
**Rating:** 3
**Confidence:** 5

**Summary:**

Summary: Large Language Models (LLMs) have become increasingly powerful and are capable of generating human-like text. This capability has raised concerns that LLMs could be used to generate misinformation. The authors investigate the difficulty of detecting LLM-generated misinformation compared with human-written misinformation. The authors find that LLM-generated misinformation can be harder to detect for both humans and detectors.

**Strengths:**

The paper is a well-written and informative contribution to the field of misinformation research. It provides important insights into the potential for LLMs to be used to generate deceptive and harmful misinformation.
- It is one of the first papers to systematically investigate the detectability of LLM-generated misinformation.
- It creates a taxonomy and identifies three different types of LLM-generated misinformation: Hallucinated News Generation, Totally Arbitrary Generation, and Partially Arbitrary Generation.
- It evaluates the detectability of different types of LLM-generated misinformation by humans.

**Weaknesses:**

Cencern1: The study is relatively small number of evaluators and only evaluates a limited number of LLM-generated news items. This means that the findings of the study may not be generalizable to all LLM-generated news items.

Concern 2: The study does not evaluate the effectiveness of different detection methods for LLM-generated misinformation. This means that it is not clear how well existing detection methods would perform at detecting the LLM-generated news items used in the study.

Jiameng Pu, Zain Sarwar, Sifat Muhammad Abdullah, Abdullah Rehman, Yoonjin Kim, Parantapa Bhattacharya, Mobin Javed, and Bimal Viswanath. Deepfake text detection: Limitations and opportunities. In 2023 IEEE Symposium on Security and Privacy (SP), pages 1613–1630. IEEE, 2023

Mitchell E, Lee Y, Khazatsky A, Manning CD, Finn C. Detectgpt: Zero-shot machine-generated text detection using probability curvature. arXiv preprint arXiv:2301.11305. 2023 Jan 26 (ICML 2023)

**Questions:**

Cencern1: The study is relatively small and only evaluates a limited number of LLM-generated news items. This means that the findings of the study may not be generalizable to all LLM-generated news items.

Concern 2: The study does not evaluate the effectiveness of different detection methods for LLM-generated misinformation. This means that it is not clear how well existing detection methods would perform at detecting the LLM-generated news items used in the study.

**Details Of Ethics Concerns:**

It requires IRB due to human evaluators.

---

> ### Author Response · Authors · 2023-11-23
> **Response to Reviewer oGY2**
>
> We greatly appreciate the feedback.
>
> 1. Cencern1: The study is relatively small number of evaluators and only evaluates a limited number of LLM-generated news items. This means that the findings of the study may not be generalizable to all LLM-generated news items.
>
> We conduct a much more extensive investigation on misinformation generated by ChatGPT, Llama-7b-chat-generated misinformation (or 13b, 70b) and Vicuna-7b-generated misinformation (or 13b, 33b) to further validate our findings. For each LLM as the misinformation generator, we compare the detection difficulty of different misinformation generation methods including Paraphrase Generation, Rewriting Generation, Open-ended Generation and human-written misinformation. We include multiple human-written misinformation datasets including Politifact, Gossipcop and CoAID. The results are in Appendix A.
>
> 2. Concern 2: The study does not evaluate the effectiveness of different detection methods for LLM-generated misinformation. This means that it is not clear how well existing detection methods would perform at detecting the LLM-generated news items used in the study.
>
> We would like to explain why we adopt the zero-shot LLM detectors (e.g., GPT-4) to evaluate the detection difficulty of LLM-generated misinformation: (1) **The zero-shot setting can better reflect the real-world scenarios of detecting LLM-generated misinformation.** Conventional supervisedly trained detectors  (such as BERT in the pre-LLM era) are likely to be impractical to detect LLM-generated misinformation in the practices for the following two reasons. First, it is difficult to obtain factuality supervision labels to train detectors for LLM-generated misinformation since it is harder for humans to detect than human-written misinformation (Section 5). Second, malicious users can easily utilize methods shown in Table 1 and close-sourced LLMs (e.g., ChatGPT) or open-sourced LLMs (e.g., Llama2) to generate misinformation at scale in different domains, types, and errors, which is hard for conventional supervisedly trained detectors to maintain effective. (2) **Zero-shot Detectors can better reflect the intrinsic detection difficulty of  LLM-generated misinformation compared with human-written misinformation.** Conventional supervisedly trained detectors (such as BERT in the pre-LLM era) are strongly influenced by the labeled training set construction and cannot reflect the detection difficulty of test set well. For example, various factors in  the labeled training set construction (including adopting human-written or LLM-generated misinformation or their combination as the training set, mixing the misinformation data with different factual information data, the domains of training set, the percentage of misinformation data in the training set) can lead to completely different detection performances of human-written or LLM-generated misinformation test set. Zero-shot Detectors, which do not need labeled training set, can better reflect the intrinsic detection difficulty of human-written or LLM-generated misinformation test set. (3) **Zero-shot LLM detectors (e.g., GPT-4) are already almost the state-of-the-art misinformation detectors.** There are already many works that have shown directly prompting LLMs such as GPT-4 in a zero-shot way can outperform conventional supervisedly trained models such as BERT on detecting human-written misinformation [1,2,3,4,5].
>
> Thus, we adopt ChatGPT-3.5, GPT-4 and Llama2 with standard prompting and Chain-of-thought Prompting strategies as the **representative detectors** to evaluate the detection difficulty of LLM-generated misinformation data and also compare it with human-written misinformation data.
>
> [1] A Multitask, Multi-lingual, Multimodal Evaluation of ChatGPT on Reasoning, Hallucination, and Interactivity. https://arxiv.org/abs/2302.04023
>
> [2] Towards Reliable Misinformation Mitigation: Generalization, Uncertainty, and GPT-4 https://arxiv.org/abs/2305.14928
>
> [3] Interpretable Unified Language Checking https://arxiv.org/abs/2304.03728
>
> [4] Fact-Checking Complex Claims with Program-Guided Reasoning ACL 2023 https://aclanthology.org/2023.acl-long.386/
>
> [5] Assessing the Effectiveness of GPT-3 in Detecting False Political Statements: A Case Study on the LIAR Dataset https://arxiv.org/abs/2306.08190
>
>
> We hope we have addressed all your concerns. Thanks.

---

### Official Review · Reviewer_jt9v · 2023-11-10

**Soundness:** 2 fair
**Presentation:** 2 fair
**Contribution:** 1 poor
**Rating:** 3
**Confidence:** 3

**Summary:**

The paper primarily discusses the ways in which LLMs can generate or can be leveraged to generate misinformation. It discusses its implications via various means of generation (established through an LLM generated misinformation taxonomy) and how easy/difficult it is detect this misinformation when compared to human-written misinformation.

The paper notes that LLM generated misinformation is harder to detect and can potentially cause more harm, through human evaluations and LLM based detection experiments.

**Strengths:**

The paper does a good job at describing the problem statement and their contributions. It's a good survey on the related techniques within this space.
- The misinformation taxonomy and the generation strategies of hallucination, Arbitrary Misinformation and Controllable Misinformation generation are interesting to note
- Utilizing CoT and non CoT prompting to study LLM based misinformation detection is interesting

Overall the paper is a comprehensive study on LLM generated misinformation and related techniques.

**Weaknesses:**

The paper lacks a review or comparison with pre-LLM era misinformation or fake news detection strategies. There are techniques within fact-finding and source-attribution space which can be leveraged to detect misinformation and those haven't been discussed.

The paper often uses Appendix sections to support the claims made which makes it less readable and less self-contained.

The paper establishes what 'detectors' are, rather late.

Overall the paper is a comprehensive study on LLM generated misinformation and related techniques, but found it to be lacking in making a significant/original innovation.

Minor:
spelling mistake in word 'Appendx' in section 4

**Questions:**

"against HC method" in section 3, page 4, is it supposed to be HG?

**Details Of Ethics Concerns:**

-

---

> ### Author Response · Authors · 2023-11-23
> **Response to Reviewer jt9v**
>
> We greatly appreciate the feedback.
>
>
> 1.  The paper lacks a review or comparison with pre-LLM era misinformation or fake news detection strategies. There are techniques within fact-finding and source-attribution space which can be leveraged to detect misinformation and those haven't been discussed.
>
>
> We would like to explain why we adopt the zero-shot LLM detectors (e.g., GPT-4) to evaluate the detection difficulty of LLM-generated misinformation: (1) **The zero-shot setting can better reflect the real-world scenarios of detecting LLM-generated misinformation.** Conventional supervisedly trained detectors  (such as BERT in the pre-LLM era) are likely to be impractical to detect LLM-generated misinformation in the practices for the following two reasons. First, it is difficult to obtain factuality supervision labels to train detectors for LLM-generated misinformation since it is harder for humans to detect than human-written misinformation (Section 5). Second, malicious users can easily utilize methods shown in Table 1 and close-sourced LLMs (e.g., ChatGPT) or open-sourced LLMs (e.g., Llama2) to generate misinformation at scale in different domains, types, and errors, which is hard for conventional supervisedly trained detectors to maintain effective. (2) **Zero-shot Detectors can better reflect the intrinsic detection difficulty of  LLM-generated misinformation compared with human-written misinformation.** Conventional supervisedly trained detectors (such as BERT in the pre-LLM era) are strongly influenced by the labeled training set construction and cannot reflect the detection difficulty of test set well. For example, various factors in  the labeled training set construction (including adopting human-written or LLM-generated misinformation or their combination as the training set, mixing the misinformation data with different factual information data, the domains of training set, the percentage of misinformation data in the training set) can lead to completely different detection performances of human-written or LLM-generated misinformation test set. Zero-shot Detectors, which do not need labeled training set, can better reflect the intrinsic detection difficulty of human-written or LLM-generated misinformation test set. (3) **Zero-shot LLM detectors (e.g., GPT-4) are already almost the state-of-the-art misinformation detectors.** There are already many works that have shown directly prompting LLMs such as GPT-4 in a zero-shot way can outperform conventional supervisedly trained models such as BERT on detecting human-written misinformation [1,2,3,4,5]
>
> Thus, we adopt ChatGPT-3.5, GPT-4 and Llama2 with standard prompting and Chain-of-thought Prompting strategies as the **representative detectors** to evaluate the detection difficulty of LLM-generated misinformation data and also compare it with human-written misinformation data.
>
>
> [1] A Multitask, Multi-lingual, Multimodal Evaluation of ChatGPT on Reasoning, Hallucination, and Interactivity. https://arxiv.org/abs/2302.04023
>
> [2] Towards Reliable Misinformation Mitigation: Generalization, Uncertainty, and GPT-4 https://arxiv.org/abs/2305.14928
>
> [3] Interpretable Unified Language Checking https://arxiv.org/abs/2304.03728
>
> [4] Fact-Checking Complex Claims with Program-Guided Reasoning ACL 2023 https://aclanthology.org/2023.acl-long.386/
>
> [5] Assessing the Effectiveness of GPT-3 in Detecting False Political Statements: A Case Study on the LIAR Dataset https://arxiv.org/abs/2306.08190

---

> ### Author Response · Authors · 2023-11-23
> **Response to Reviewer jt9v (2)**
>
> 2. Overall the paper is a comprehensive study on LLM generated misinformation and related techniques, but found it to be lacking in making a significant/original innovation.
>
> We would like to emphasize the major contributions summarized in the introduction part as follows:
>
> > 1. We build the first taxonomy by types, domains, sources, intents and errors to characterize LLM- generated misinformation as an emerging research topic.
> > 2. We are the first to categorize and validate the potential real-world methods for generating misinformation with LLMs including Hallucination Generation, Arbitrary Misinformation Generation and Controllable Misinformation Generation.
> > 3. We discover that misinformation generated by LLMs (e.g., ChatGPT) can be harder for humans and detectors to detect than human-written misinformation with the same semantic information, which provides empirical evidence to demonstrate that LLM-generated misinformation can be more deceptive and potentially cause more harm.
> > 4. We discuss the emerging challenges for misinformation detectors (Section 6), implications of our discovery on combating misinformation in the age of LLMs (Section 7) and the countermeasures through LLMs’ lifecycle (Section 8).
>
> And our non-trivial discovery “LLM-generated misinformation can be harder to detect for humans and detectors compared to human-written misinformation with the same semantics" have multiple important implications for combating misinformation in the age of LLMs (in Section 7).
>
> > 1. LLM-generated misinformation can be more deceptive and potentially cause more harm.
> > 2. There is a potential major paradigm shift of misinformation production from humans to LLMs.
> > 3. Collective efforts on combating LLM-generated misinformation from different stakeholders are needed.
>
> 3. The paper often uses Appendix sections to support the claims made which makes it less readable and less self-contained. The paper establishes what 'detectors' are, rather late.
>
> We will improve on these aspects in our future revision.
>
> 4. "against HC method" in section 3, page 4, is it supposed to be HG? spelling mistake in word 'Appendx' in section 4
>
> We have revised the typos.
>
> We hope we have addressed all your concerns. Thanks.

---

### Author Response · Authors · 2023-11-23
**General response**

Dear reviewers,

We are grateful for the constructive feedbacks. We have improved our paper in following aspects:

1. We conduct a much more extensive investigation on misinformation generated by ChatGPT, Llama-7b-chat-generated misinformation (or 13b, 70b) and Vicuna-7b-generated misinformation (or 13b, 33b) to further validate our findings. For each LLM as the misinformation generator, we compare the detection difficulty of different misinformation generation methods including Paraphrase Generation, Rewriting Generation, Open-ended Generation and human-written misinformation. We include multiple human-written misinformation datasets including Politifact, Gossipcop and CoAID. The results are in Appendix A.
2. We release our whole datasets and detection results for reproduction and verification. https://anonymous.4open.science/r/llm-misinformation-iclr2024-7E35/
3. We carefully discuss the statistical significance of all the results in Appendix B.
4. We illustrate the reasons for our evaluation setting more clearly.

We have updated the submission pdf document to include our revision. **All revision text is marked with blue.**

We believe we have addressed all the concerns. Thanks.

---

### Meta-Review · Area_Chair_xxhk · 2023-12-20

**Metareview:**

The paper focuses on exploiting LLMs to generate misinformation. This is an impactful direction from many aspects including the societal aspect. The contributions of the paper are as follows: First, thee asuthors ask if LLM-generated misinformation can cause more harm than human-written misinformation? They propose to tackle this question from the perspective of detection difficulty by building a taxonomy of LLM-generated misinformation.  Second, the paper provides a comprehensive study, indicating that LLM-generated misinformation can be harder to detect for humans and detectors compared to human-written misinformation with the same semantics, which suggests it can have more deceptive styles and potentially cause more harm.

The reviewer's opinion about this paper was mixed. After the rebuttal, two reviewers (who have worked quite a bit on the topic of mis-information) indicated that they found the results interesting. Reviewer NKW5 also increased their score to 6 (although they were not able to enter it in open review). Unfortunately, the other two reviewers who gave a score 3 were not responsive despite my many messages on openreview (and private messages by email). What the reviewers (and myself) liked about the paper is the compressive study and the human evaluation.

Overall, I decide to accept the paper as two reviewers (and myself) are in favor of accepting it. I strongly encourage the authors to incorporate all the great comments from the reviewers in the revised version.

**Justification For Why Not Higher Score:**

See above

**Justification For Why Not Lower Score:**

See above

---

### Decision · Program_Chairs · 2024-01-16

Accept (poster)